# A Retrospective Evaluation of Self-Reported Adverse Events Following Immunization with Different COVID-19 Vaccines in Türkiye

**DOI:** 10.3390/vaccines11020316

**Published:** 2023-01-31

**Authors:** Sultan Mehtap Büyüker, Arifa Sultana, Jakir Ahmed Chowdhury, Abu Asad Chowdhury, Shaila Kabir, Md. Shah Amran

**Affiliations:** 1Department of Pharmacy Services, Üsküdar University, 34662 İstanbul, Turkey; 2Molecular Pharmacology and Herbal Drug Research Laboratory, Department of Pharmaceutical Chemistry, Faculty of Pharmacy, University of Dhaka, Dhaka 1000, Bangladesh; 3Department of Pharmaceutical Technology, Faculty of Pharmacy, University of Dhaka, Dhaka 1000, Bangladesh

**Keywords:** pharmacovigilance, side-effects, COVID-19, adverse events following immunization, corona vaccines, cross-sectional study, Türkiye

## Abstract

Background: The Sinovac and BioNTech vaccines were the first to be introduced in Türkiye to fight the ongoing global COVID-19 pandemic. As these vaccines had shown some side-effects in its clinical trial, we aimed to conduct a survey study to assess the short-term adverse events following immunization (AEFIs) in Türkiye. Method: A cross-sectional study was conducted using social and electronic media platforms by delivering a pre-formed and validated online questionnaire among people who had received at least one dose of the COVID-19 vaccine. This survey study focused on mass populations from different regions in Türkiye. A total of 603 responses were collected. Among these, 602 were selected based on complete answers and used for the assessment. The collected data were then analyzed to evaluate the various parameters related to the AEFIs of the respondents. Results: Among the total 602 participants, 20.8% were male, and 78.7% were female, actively answering all of the constructive questions. Most of the respondents were between 18–30 years of age. We found that a total of 23.3% of the total respondents had been infected with the SARS-CoV-2 virus. Our survey revealed that out of 602 volunteers, the rate of experiencing physical discomfort was higher in participants who had received the Pfizer-BioNTech vaccine at all three doses than in those who had received the Sinovac vaccine. When all vaccine types were examined, the most common side effect was pain at the injection site, reported by 75.19% participants. When the side effects were compared according to vaccine types, there was a significant difference only in terms of fever. Fever rates in those who had received the Pfizer-BioNTech vaccine (20.96%) were found to be significantly higher than those who had received the Sinovac vaccine (8%). Conclusions: The studied vaccines showed minor side effects and there was no significant difference between the vaccines in terms of other side effects. Moreover, further research is needed to determine the efficacy of the existing vaccines in preventing SARS-CoV-2 infections or after-infection hospitalization.

## 1. Introduction

The COVID-19 pandemic has emerged as a serious public health emergency and has triggered a process that needs to be responded to. On 11 March 2020, the World Health Organization (WHO) declared this event a public health emergency in accordance with the International Health Regulations. SARS-CoV-2 caused a pandemic as a virus that has been encountered since the 1918 flu epidemic and the smallpox pandemic and has caused serious illness [1,2]. In the second week of December 2019, it emerged in Wuhan, Hubei province of China, and patients were diagnosed with atypical pneumonia [3] and a new corona virus named nCoV-2019 was detected. In mid-January of 2020, the virus was definitively identified and announced to the public as SARS-CoV-2, hence the name COVID-19. This virus has spread rapidly in China, where it originated, and then all over the world. On 11 March, the World Health Organization (WHO) declared this situation as a pandemic all over the World [4,5,6,7,8]. As of 2 July 2021, the WHO has detected more than 182 million cases of COVID-19 worldwide, and more than three-million deaths have occurred. The emergence of the COVID-19 mutation and the global health problem it created enabled the formulation of effective and safe vaccines for the emerging deadly variants [9]. The growing global epidemic has not only caused health problems, but has also triggered many economic and social problems [10,11].

Social restrictions have been introduced at certain times in order to control the spread of the increasing number of cases throughout Türkiye [12,13]. In order to control the epidemic in Türkiye, as in the whole world, it aimed to control the pandemic by vaccinating more than 70% of the population with safe, effective, cost-effective and accessible vaccines [14]. With the epidemic, COVID-19 vaccine studies started in Türkiye, and drug and vaccine development projects for the treatment of COVID-19 continue [15]. Phase-III studies were carried out on the inactivated SARS-CoV-2 vaccine developed by China in mid-September 2020. The vaccination practices continued in line with the strategies determined by the Ministry of Health of Türkiye [16]. This process is closely followed by the Health Ministry of Türkiye, as well as by the rest of the world. One of the interventions planned to be carried out by the Ministry in order to respond to the COVID-19 pandemic is the massive COVID-19 vaccination program. The groups to be vaccinated against COVID-19 have been determined by evaluating the risks of exposure, severe transmission of the disease, and the negative impact of the disease on the functioning of social life, and these groups were given the preference in administering the vaccines [17].

After the limited-dose vaccine administration, the priority of the immunization program, i.e., who will be given the vaccine first, and the community demand for vaccines are discussed. For the first time in Türkiye, an agreement was made with fifty-million doses of inactivated vaccines for vaccination, and after the first three-million doses were obtained, emergency use permission was obtained and it was intended to vaccinate all eligible persons over the age of 18, primarily healthcare workers, and the vaccination process began. The order of the vaccine administration was determined by the Ministry of Health, as shown in Table A1 [17].

According to some studies, the Oxford-AstraZeneca vaccine is less efficient against the virulent B.351 strain that has emerged in South Africa [18]. Furthermore, researchers uncovered side effects of the Pfizer vaccine, noting that it can cause arrhythmia, myocarditis, and even death in some people. In Israel, two people died among 62 male patients who received the Pfizer vaccine [19]. In addition, in the UK, AstraZeneca reported post-vaccination infection rates of 0.3 per cent; for Pfizer, it was 0.8 per cent [20]. In the US, the Moderna vaccine resulted in higher adverse responses in individuals who received the vaccine from a specific vaccine center in California. As a result, the vaccination program at that center was halted [21]. Due to these factors, post-vaccination surveillance is necessary at this point of the vaccination rollout in Türkiye in order to increase public confidence and evaluate the actual efficacy and safety of the licensed vaccines. The results of this study will be reassuring to those who are uncertain about different COVID-19 vaccines. Therefore, the primary aim of this study was to provide evidence of the different COVID-19 vaccine side effects after receiving four doses delivered in Türkiye. The other major objectives were to assess the perception of people toward the COVID-19 vaccine and find the association between different side effects and various demographic and clinical characteristics.

## 2. Materials and Methods

### 2.1. Design and Sample Selection

This survey study on AEFIs of the COVID-19 vaccines was performed online using a retrospective and cross-sectional method. We prepared a survey questionnaire after a careful review of the COVID-19 data and surveillance from the Centers for Disease Control and Prevention (CDC) [22]. An extensive literature review on the associated side-effects of the COVID-19 vaccines [9] [23,24,25,26,27,28,29,30,31,32] and a group discussion was integrated to finalize the questionnaire. Ethical approval was obtained from the human ethical review committee of the University of Üsküdar to conduct the study (approval number: 61351342/April 2022-38). The online questionnaire was then distributed over social and electronic media (Email, Facebook, Twitter, and WhatsApp) using a snowball sampling method.

The participants were encouraged to forward the questionnaire link to others in their social networks. The intended participants were Turkish individuals 18 years of age and older, and who could read and interpret Türkish or English. Due to the constraints of utilizing face-to-face techniques during an active outbreak, the data were solely collected using the Google Forms platform. This online form was extensively shared via social and electronic sites in Türkiye and was widely utilized by people of all socioeconomic backgrounds and different age groups.

### 2.2. Questionnaire Preparation

The questionnaire was created in response to the circumstance through group discussion. The survey form consisted of seven sections, containing vaccination information, health condition before and after the vaccination, associated side-effects after the vaccination, any symptom management step taken by participants, etc. The first section contained the general information about this survey and asked for consent. All of the respondents were obligated to answer this section in order to continue with the survey. The second section contained personal information such as age, gender, residence, educational qualification, etc. The next section contained a question concerning the vaccination information of the current individual, including vaccine name, vaccination date, dose, etc. The fourth section was specifically designed for females. It contained three questions, including the pregnancy and lactation condition of the female. In the initial stage of COVID-19 pandemic, there was a rumor that those who have received the tetanus vaccine are less susceptible to the COVID-19 virus. For this reason, the tetanus vaccination status was also included in this section. The following section presented several questions related to the current health status of the individuals, before the vaccination. This section contained questions regarding the current COVID-19 status of the participant and preventive measures taken, such as pneumonia vaccinations or plasma therapy. This section also addressed allergic conditions, chronic diseases with current treatment patterns, previous vaccination information, etc. Most of the questions in this section were in a dichotomous ‘yes’ or ‘no’ format. Section six of the questionnaire was headlined as “After Effects Following Vaccination” and contained only two questions: (a) Had the participant been affected by COVID-19 before the first dose? and (b) Did the participant face any physical discomfort? Both were in the dichotomous ‘yes’ or ‘no’ format. The seventh section was only for respondents who had responded “yes” to the last question in the previous section. This section presented the type, duration, management, and treatment pattern of physical discomfort after the vaccination. The original questionnaire was prepared in English but later translated into Turkish for easy understanding.

### 2.3. Duration of the Study

The study was conducted between 1 September and 30 November 2022. A response period of 12 weeks was allocated in order to collect the replies from the COVID-19 vaccine recipients.

### 2.4. Statistical Analysis

Following the data collection stage, the data set was reviewed. Individuals who participated in the study even though they had not been vaccinated were excluded from the study. The frequency and percentage distributions of the demographic data of the participants were examined on the data set. In addition, questions about the vaccination status of participants, symptoms observed after vaccination, hospitalization status, and the treatments given were examined for their frequency and percentage distributions. Chi-square analysis was performed to compare the physical discomfort and side effects seen in participants, according to vaccine types. The Monte Carlo p-value was used in case of the presence of cells with a frequency value of less than five observed in the chi-square analysis. The analyses were conducted on the SPSS 22 software package, and the level of significance was determined as α = 0.05.

## 3. Results

This study involved 603 volunteers from various socioeconomic backgrounds who had received at least one vaccination dose. In the end, 602 responses with complete answers were selected for the final analysis.

Table 1 illustrates the demographic data of the participants. It was observed that, around three-quarters of the total participants, more than half of them, were female (78.7%) and teenagers (75.9%). There were also two respondents above 70 years old. The majority of the respondents live in province areas (92.5%), but there were also respondents from rural areas (6.3%) and from abroad (1.2%). In total, 260 (43.2%) respondents were pursuing their undergraduate degree, while the other 242 (40.2%) were completing their associate degree. Only one respondent was from middle school and four respondents were from elementary school.

Figure 1 shows the participants’ beliefs about whether vaccines can prevent COVID-19 infection. The majority of the participants (63.3%) responded “agree” or “strongly agree” to the statement that the vaccines had a preventive function.

Table 2 shows the distributions of the status of pregnancy and breastfeeding after vaccination. While two of the 474 women (0.4%) were vaccinated while pregnant, one (50%) of these women experienced an abnormality during pregnancy. Of the women, 1.5% breastfed their child after they obtained the vaccine. In addition, 52.3% of all participants had previously received a tetanus vaccine.

In addition, 23.3% of the participants were infected by COVID-19 before vaccination. The majority of these individuals (74.3%) were given medication treatment, which was followed by herbal support with 50%, treatment in the hospital with 17.1%, and plasma treatment with 2.1%. These are shown in Table A2. The rate of hospitalization after participants contracted COVID-19 was 2.5%. These individuals were mostly diagnosed with upper respiratory tract infection (60%), low oxygen saturation (53.3%), and lung infection (26.7%). These are shown in Table A3. The participants were infected for between 7 and 15 days, with a highest infection rate of 51.19%.

While the majority of the participants (79.9%) did not experience COVID-19-like symptoms before being vaccinated, 8.5% did. In addition, 10.8% of the participants were suffering from chronic diseases and 18.1% had allergies. While 26.1% of participants had received a vaccine for another disease, 9.1% had been vaccinated against pneumonia and/or flu for preventive purposes.

Table 3 shows the distribution of the vaccination rates and vaccine types. Among the 602 participants who had received the first dose of the vaccine, 93.2% obtained the second dose, 10.1% obtained the third dose, and 3.5% obtained the fourth dose. The Pfizer-BioNTech was the most commonly received vaccine type. Additionally, 78.4% of the first dose vaccines, 78.4% of the second dose vaccines, 80.3% of the third dose vaccines, and all of the fourth dose vaccines were of the Pfizer-BioNTech vaccine followed by the Sinovac vaccine.

Table 4 shows the Chi-square analysis conducted for the comparison of the physical discomfort experienced after being vaccinated. According to the results of the analysis, a significant difference was found between vaccine types in terms of experiencing physical discomfort after the first dose (χ^2^ = 55.773, *p* ˂ 0.001), the second dose (χ^2^ = 66.311, *p* ˂ 0.001), and the third dose (χ^2^ = 7.958, *p* ˂ 0.01). The rate of experiencing physical discomfort was higher in participants who had received the Pfizer-BioNTech vaccine at all three doses than in those who had received the Sinovac vaccine.

Table 5 shows the vaccine types and the distribution of the side effects with each dose. For the first dose, all vaccine types were examined, and the most common side effect was pain at the injection site, at 75.19%. When the side effects were compared according to vaccine types, there was a significant difference only in terms of fever symptoms (χ^2^ = 4.715, *p* ˂ 0.05). The fever rates seen in those who had received the Pfizer-BioNTech vaccine (20.96%) were found to be significantly higher than the rates of fever seen in those who had received the Sinovac vaccine (8%). There was no significant difference between the vaccines in terms of the other side effects. After taking second dose, the most common side effect was pain at the injection site, and it was 76.99%. When the side effects were compared according to vaccine types, there was a significant difference in terms of body pain (χ^2^ = 4.825, *p* ˂ 0.05), joint pain (χ^2^ = 5.92, *p* ˂ 0.05), and nausea (χ^2^ = 5.311, *p* ˂ 0.05) symptoms. The rates of body pain (44.12%), joint pain (32.68%), and nausea (14.05%) seen in participants who had received the Pfizer-BioNTech vaccine were higher than the rates of body pain (24.24%), joint pain (12.12%), and nausea (0%) seen in Sinovac vaccine recipients. In the case of the third dose, the most common side effect was pain at the injection site, and it was 83.78%. However, there was no significant difference between the vaccine types in terms of side effects (*p* ˃ 0.05) for the third dose. The type of vaccine received by all 13 participants who felt physical discomfort after the fourth dose was the Pfizer-BioNTech vaccine. Among these participants, 92.31% had pain at the injection site, 53.85% had body pain, and 38.46% had joint pain and headache.

Figure 2 illustrates that the prevalence of side-effects appeared in a similar pattern after taking each of the four doses.

## 4. Discussion

In this study, we assessed the self-reported adverse events observed by COVID-19 vaccine recipients in Türkiye. Before 30 November 2022, four doses of COVID-19 were administered in Türkiye [33]. A total of 603 responses were obtained via an online questionnaire, including those who took at least one dose of the COVID-19 vaccine. Among them, 602 respondents were selected for analysis based on their complete responses. Table 1 presents the demographic characteristics of the respondents. Among the 602 participants, 474 (78.7%) respondents were female and 457 (75.9%) of the total respondents were in the 18–30 age range. The majority of the respondents (92.5%) were from the province area. Among the total respondents, 43.2% were from the undergraduate level and 40.2% were taking an associate degree.

Figure 1 illustrates that the majority of the respondents agreed with the statement, “Administering COVID-19 vaccine can reduce the severity of subsequent infection”. Among the 602 participants, 381 (63.3%) respondents agreed with the statement. Table 2 illustrates the particular physical condition of the female respondents. The rate of receiving COVID-19 vaccines in pregnant women was much lower than normal, because of the insecurity and fear towards the vaccine [34]. It was observed that 2 of the 474 women (0.4%) received COVID-19 vaccines when they were pregnant. Among them, one woman faced some difficulties during her pregnancy, which may or may not be related to the COVID-19 vaccines. In another study conducted among health workers, a similar pattern was observed in adverse events following vaccination in pregnant and nonpregnant women. In addition, any pregnancy-related adverse events were rarely reported [35]. Among the participants, seven respondents were lactating mothers at the time of receiving the vaccination. More than half of the female participants previously received the tetanus vaccine that was supposed to play a role in reducing the severity of corona [36]. Approximately 140 participants were infected by COVID-19 before the vaccination program started. Most of them had a duration of 7–15 days of infection. Among them, half of the people obtained herbal therapy, most of the people took medicine with or without a doctor’s advice, 24 of them were admitted to the hospital, and three patients received plasma therapy to treat the infection. The hospitalized patients were admitted to the hospital mostly for upper respiratory tract infections, low oxygen saturation, and lung infection. Some other studies also support the data [37,38]. Few of the participants (8.5%) felt COVID-like symptoms before vaccination and 9.1% of participants received pneumonia vaccines to prevent COVID-19 infection before the COVID-19 vaccines were available.

Prior to 30 November 2022, 602 participants received the first dose, 561 received the second dose, 61 received the third dose, and 21 received the fourth dose of the COVID-19 vaccine. Among them, the majority received the Pfizer-BioNTech vaccine followed by the Sinovac vaccine. The number of participants who received Moderna, Sputnik V, and Turkovac vaccines can be overlooked. Among the total participants, 353 of the 472 Pfizer-BioNTech vaccine receivers and 50 of the 126 Sinovac receivers felt some kind of physical discomfort after receiving the first dose of the vaccine. The rate of reporting physical discomfort after receiving the vaccines was greater in the Pfizer-BioNTech vaccine receivers than that of the Sinovac vaccine receivers for the second and third doses also.

Pain at the site of injection was the most common discomfort felt after each dose of the COVID-19 vaccines, followed by joint pain, arm numbness, and tiredness. Pain at the injection site and fever were dominant among the other side effects after receiving the first dose of the COVID-19 vaccines; pain at the injection site, body pain, joint pain, and nausea were prominent after the second dose; pain at the injection site was prominent after the third dose; and pain at the injection site, body pain, joint pain and headache were prominent after taking the fourth dose, which was solely for the Pfizer-BioNTech vaccine. In our previous study, conducted in Bangladesh, the common side effects reported after receiving the COVID-19 vaccines included swelling and pain at the injection site and fever [39]; pain at the injection site, fatigue, headache, muscle pain, and chills were the most commonly reported side effects found in a Czech Republic-based survey study after receiving the Pfizer-BioNTech COVID-19 vaccine [40]. Another study, from Saudi Arabia, illustrated that the side effects associated with the Pfizer-BioNTech and Oxford-AstraZeneca vaccines included fatigue and pain at the site of the injections as major reported side effects [26]. The type and prevalence of the side effects varied extensively according to the country and type of vaccines [21,26,40,41,42,43,44,45,46,47]. Moreover, it was observed that different types of pain were predominant in all of the vaccine types. Lanitis et al. showed that the educational qualification of a person is highly related to the feeling of pain [48]. The less educated person feels more pain than the educated person. However, in our study, all of the participants were at least at their undergraduate level; thus, educational qualification did not affect the pain or other physical discomforts, rather, it varied according to vaccine types. Additionally, it is worth noticing that the percentage of physical discomforts showed similar patterns after all four doses, although the vaccine distribution was not same.

### 4.1. Strengths of the Study

The major strength of this study is that the sample was drawn from the educated people of Türkiye, who are anticipated to offer reliable information based on their conditions. Moreover, we only included the side effects that can be identified by common people so that the respondents do not misinterpret the data. According to our understanding, this is the first independent research to examine the adverse effects of all four doses of the COVID-19 vaccines.

### 4.2. Limitations of the Study

Our study included a number of limitations, despite the fact that we conducted a thorough analysis of the data to arrive at a valid conclusion. We conducted an online survey and were unable to acquire enormous amounts of data through face-to-face communication. We evaluated the responses based completely on trust rather than verified investigations by healthcare professionals. In addition, the response rate could not be estimated because the number of vaccinated persons in Türkiye was rising at the time this study was conducted. In our study, we solely analyzed the acute, short-term adverse consequences of the vaccines. Consequently, it is vital to evaluate the long-term side effects of the ongoing immunization programs.

## 5. Conclusions

COVID-19 severity is substantially decreased with the vaccine rollout conducted periodically in Türkiye. Minor adverse effects were reported with the test vaccines, and there was no discernible variation in the side effects between vaccines. The side effects are also decreasing with each dose of the COVID-19 vaccine received. However, further investigation is required to establish the effectiveness of the currently available vaccines that has the least side effects to prevent SARS-CoV-2 infections.

## Figures and Tables

**Figure 1 vaccines-11-00316-f001:**
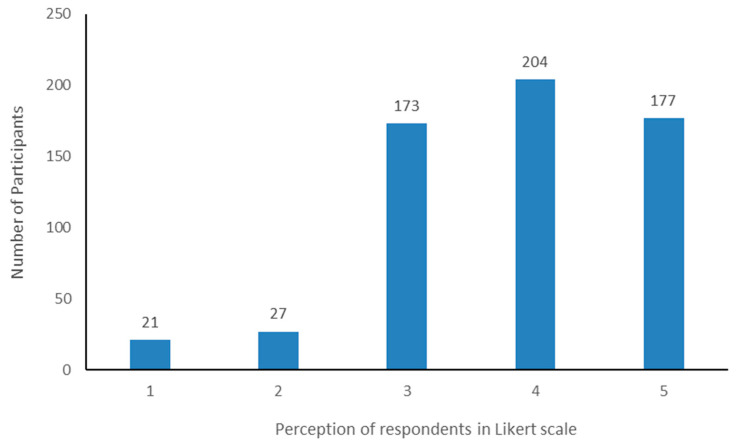
Perception of respondents toward the question “Can vaccines prevent COVID-19 infection?” in Likert scale (1: strongly disagree; 2: disagree; 3: neutral; 4: agree; and 5: strongly agree).

**Figure 2 vaccines-11-00316-f002:**
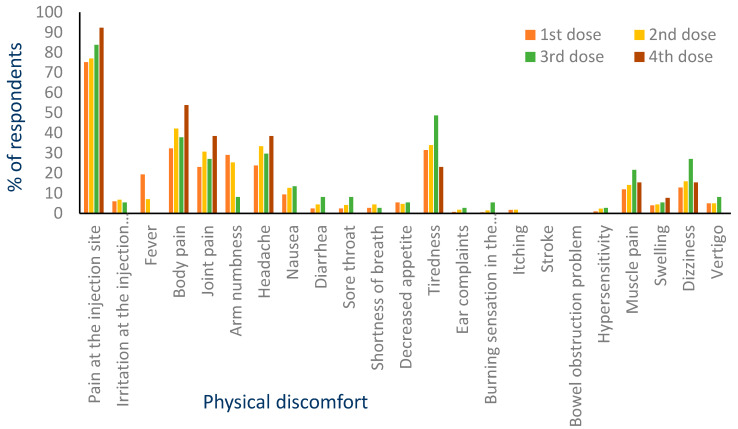
Distribution of side effects according to doses.

**Table 1 vaccines-11-00316-t001:** Demographic data of participants from a cross-sectional study in Türkiye (*n* = 602).

Characteristics	Categories	Frequency	Percentage (%)
Gender	Male	125	20.8
	Female	474	78.7
	Not stated	3	0.5
Age	<18	29	4.8
	18–30	457	75.9
	31–40	38	6.3
	41–50	53	8.8
	51–60	20	3.3
	61–70	3	0.5
	>70	2	0.3
Place of residence	Rural area	38	6.3
	Province	557	92.5
	Abroad	7	1.2
Education	Elementary school	4	0.7
	Middle school	1	0.2
	High school	53	8.8
	Undergraduate degree	260	43.2
	Associate degree	242	40.2
	Master’s degree	31	5.1
	PhD	11	1.8

**Table 2 vaccines-11-00316-t002:** Pregnancy and breastfeeding status after vaccination.

Parameters	Outcome	Frequency (f)	Percentage (%)
Status of pregnancy after vaccination	No	472	99.6
	Yes	2	0.4
Abnormality during pregnancy	No	1	50.0
	Yes	1	50.0
Breastfeeding after vaccination	No	467	98.5
	Yes	7	1.5
Having a tetanus vaccine	No	287	47.7
	Yes	315	52.3

**Table 3 vaccines-11-00316-t003:** Vaccination rates and types of vaccines.

	First Dose		Second Dose		Third Dose		Fourth Dose	
	f	%	f	%	f	%	f	%
Getting vaccinated								
Yes	602	100	561	93.2	61	10.1	21	3.5
No	0	0	41	6.8	541	89.9	581	96.5
The vaccines								
Moderna	2	0.3	2	0.4	0	0	0	0
Pfizer-BioNTech	472	78.4	440	78.4	49	80.3	21	100
Sinovac	126	20.9	117	20.9	12	19.7	0	0
Sputnik V	1	0.2	1	0.2	0	0	0	0
Turkovac	1	0.2	1	0.2	0	0	0	0
f = Frequency								

**Table 4 vaccines-11-00316-t004:** Comparison of physical discomfort experienced after being vaccinated.

Vaccine No.	Name of Vaccines	Infestation of Physical Discomfort after Vaccination		χ^2^	*p*
		Frequency	%		
The first dose	Pfizer-BioNTech	353	74.8%	55.773	0.000
	Sinovac	50	39.7%		
The second dose	Pfizer-BioNTech	306	69.5%	66.311	0.000
	Sinovac	33	28.2%		
The third dose	Pfizer-BioNTech	34	69.4%	7.958	0.008 ^a^
	Sinovac	3	25.0%		
The fourth dose	Pfizer-BioNTech	13	61.9%	-	-
	Sinovac	0	0.0%		

^a^ Monte Carlo *p* value ˂ 0.05.

**Table 5 vaccines-11-00316-t005:** Vaccine types and distribution of side effects after taking 1st, 2nd, 3rd, and 4th dose of vaccines.

Sl No.	Physical Discomfort Reported after Taking COVID-19 Vaccines	First Dose		Second Dose		Third Dose		Fourth Dose
		Pfizer-BioNTech	Sinovac	Pfizer-BioNTech	Sinovac	Pfizer-BioNTech	Sinovac	Pfizer-BioNTech
1	Pain at the injection site	265	38	236	25	28	3	12
2	Irritation at the injection site	21	3	21	2	2	0	0
3	Fever	74	4	24	0	0	0	0
4	Body pain	119	11	135	8	14	0	7
5	Joint pain	86	7	100	4	10	0	5
6	Arm numbness	107	10	78	8	3	0	0
7	Headache	84	12	106	7	10	1	5
8	Nausea	37	1	43	0	4	1	0
9	Diarrhea	8	2	14	1	2	1	0
10	Sore throat	10	0	13	1	2	1	0
11	Shortness of breath	10	1	14	1	1	0	0
12	Decreased appetite	21	1	15	1	2	0	0
13	Tiredness	111	16	107	8	18	0	3
14	Ear complaints	3	0	6	0	1	0	0
15	Burning sensation in the stomach	2	0	4	1	2	0	0
16	Itching	7	0	6	0	0	0	0
17	Stroke	0	0	0	0	0	0	0
18	Bowel obstruction problem	0	0	1	0	0	0	0
19	Hypersensitivity	4	0	8	0	1	0	0
20	Muscle pain	43	5	43	5	7	1	2
21	Swelling	15	1	13	2	1	1	1
22	Dizziness	42	10	50	4	10	0	2
23	Vertigo	19	1	16	1	2	1	0

## Data Availability

The data presented in this study are available on request from the corresponding author. The data are not publicly available due to large complicated data set.

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
