# Peer review of "A Retrospective Evaluation of Self-Reported Adverse Events Following Immunization with Different COVID-19 Vaccines in Türkiye"

_vaccines, 2023, doi:10.3390/vaccines11020316_

Round 1
Reviewer 1 Report
I think this is a well organized, well presented survey about COVID-19 vaccination predominately in young adult females. The data are well presented and clinically relevant in this group of 602 adults.
Author Response
We would like to convey our warm gratitude and thankfulness to the honourable reviewer for giving time and effort.
Reviewer 2 Report
Authors should resubmit the manuscript after correcting it in accordance with major and minor revisions.
Comments
Researchers conducted a survey study to reveal the side effects of COVID-19 vaccines in Turkey. While pain at the injection site was the most common side effect in all vaccine types, fever showed a significant difference when the side effects were compared according to the vaccine types.
Minor Reviews
· All spelling and grammatical errors should be corrected. For example, " COVİD-19" on lines 51,54, and 55.
· In the introduction part, a lot of basic information about COVID-19 is mentioned. Instead, a paragraph on the side effects of vaccines should be added in line with the purpose of the study.
· Reference 10 in the introduction is improperly constructed. I recommend deleting it.
· In line 146 and 147 ''This section may be divided by subheadings. It should provide a concise and precise description of the experimental results, their interpretation, as well as the experimental conclusions that can be drawn.'' statament is confusing and unrelated.
· The reason why the tetanus vaccination status is questioned should be stated in the method section.
Major Reviews
· Lanitis et al. showed a relationship between educational status and perception pain. It would be interesting to reveal the relationship between the educational status of the participants and their perception of pain. Thus, the state of education as a confounding factor will be checked.
· The literature should be considered more comprehensively than the results are repeated in the discussion part. The introduction and discussion parts are clumsy and should be improved.
Reference
Lanitis S, Mimigianni C, Raptis D, Sourtse G, Sgourakis G, Karaliotas C. The Impact of Educational Status on the Postoperative Perception of Pain. Korean J Pain. 2015 Oct;28(4):265-74. doi: 10.3344/kjp.2015.28.4.265. Epub 2015 Oct 2. PMID: 26495081; PMCID: PMC4610940.
Author Response
We would like to convey our warm gratitude and thankfulness to the honourable reviewer for giving time and taking the pain to assess our manuscript and come up with thoughtful and constructive comments. We tried our best to reply to all the comments and questions of the honourable reviewer.
Comments from the Second Reviewer
Minor Reviews:
- “All spelling and grammatical errors should be corrected. For example, " COVİD-19" on lines 51,54, and 55”.
Our reply:
We have corrected them according to reviewers comment as follows,
“In the second week of December 2019, it emerged in Wuhan, Hubei province of China, and patients were diagnosed with atypical pneumonia [3] and a new corona virus named nCoV-2019 was detected. In mid-January of 2019, the virus was definitively identified and announced to the public as SARS-CoV-2, hence the name COVID-19. This virus has spread rapidly in China, where it originated, and then all over the world. On March 11, the World Health Organization (WHO) declared this situation as a pandemic all over the World [4–8]. As of July 2, 2021, the WHO has detected more than 182 million cases of COVID-19 worldwide, and more than 3 million deaths have occurred. The emergence of the COVID-19 mutation and the global health problem it creates have enabled the formulation of effective and safe vaccines for the emerging deadly variants [9].”
- In the introduction part, a lot of basic information about COVID-19 is mentioned. Instead, a paragraph on the side effects of vaccines should be added in line with the purpose of the study.
Our reply:
We have added a paragraph describing the side effects of different COVID-19 vaccines in line number 82 and onward. The paragraph is as follows-
“According to some studies, the Oxford-AstraZeneca vaccine is less efficient against the virulent B.351 strain that has emerged in South Africa [18]. Furthermore, researchers uncovered Pfizer vaccine side effects, noting that it can cause arrhythmia, myocarditis, and even death in some people. In Israel, 2 people died among 62 male patients who received the Pfizer vaccine [19]. In addition, in the UK, AstraZeneca reported post-vaccination infection rates of 0.3 per cent and for Pfizer, it was 0.8 per cent [20]. In the US, the Moderna vaccine resulted in higher adverse responses in individuals who specifically received the vaccine from a specific vaccine center in California. As a result, the vaccination program at that center was halted [21].”
- “Reference 10 in the introduction is improperly constructed. I recommend deleting it.”
Our Reply:
We have removed reference 10 from the introduction part and adjusted the references accordingly.
- “In line 146 and 147 ''This section may be divided by subheadings. It should provide a concise and precise description of the experimental results, their interpretation, as well as the experimental conclusions that can be drawn.'' statement is confusing and unrelated.”
Our reply:
We apologize for the inconvenience; We have deleted these lines.
- “The reason why the tetanus vaccination status is questioned should be stated in the method section.”
Our reply:
We have included the reason in line number 130 and onward as follows-
“In the initial stage of COVID-19 pandemic, there was a rumor that those who have taken tetanus vaccine, are less susceptible toward COVID-19 virus. For this reason, the tetanus vaccination status was also included in this section.”
Major Reviews:
- “Lanitis et al. showed a relationship between educational status and perception pain. It would be interesting to reveal the relationship between the educational status of the participants and their perception of pain. Thus, the state of education as a confounding factor will be checked.”
Our reply:
We have mentioned the correlation between educational status and pain according to the suggested paper and our study in line number 315 and onward. The inserted lines are as follows-
“Moreover, it was observed that different types of pain were predominant in all vaccine types. Lanitis et al. showed that the educational qualification of a person is highly related to the feeling of pain. The less educated person feels more pain than the educated person. But in our study, all of the participants were at least at their undergraduate level, educational qualification did not affect the pain or other physical discomforts, rather it varied according to vaccine types [50].”
- “The literature should be considered more comprehensively than the results are repeated in the discussion part. The introduction and discussion parts are clumsy and should be improved.”
Our reply:
We have critically evaluated the introduction and discussion part and did necessary corrections suggested by honorable reviewers.
Reviewer 3 Report
Thank you for the opportunity to review this article which presents the results of a survey administered to the Turkish general population to investigate the adverse events after the administration of different types of anti-COVID vaccinations (self-reported). In general terms, the study has been conducted in a proper manner, and results are presented quite clearly. Also, the Introduction is informative and the Discussion is appropriate. In my opinion, the major issue about this article is the limited importance of the results, but this has to do with the design of the study, which obviously cannot be modified at this point. I would suggest the authors to emphasize this aspect and maybe stress the principal aim for which the study was made, that, from what I could understand, is to provide the general Turkish population with general information about the safety of anti-COVID vaccines.
I would suggest to re-edit some of the tables for the sake of clarity and readability, especially Tables 4 and 5.
In my opinion, Figure 2 should be completely re-edited, as the continuos line has no sense in the presentation of those results.
Please amend the start of the 'Results' section (lines 146-148).
Moderate review of English languarge is required.
Author Response
We would like to convey our warm gratitude and thankfulness to the honourable reviewer for giving time and taking the pain to assess our manuscript and come up with thoughtful and constructive comments. We tried our best to reply to all the comments of the honourable reviewer.

Round 2
Reviewer 2 Report
Reference 50 is given as a reference to the sentence of this study. This needs to be corrected.
Author Response
We would like to convey our warm gratitude to the honourable reviewer for giving time and effort to review our manuscript. We have corrected the reference and moved it to line number 314. The corrected lines are as follows-
“Moreover, it was observed that different types of pain were predominant in all vaccine types. Lanitis et al. showed that the educational qualification of a person is highly related to the feeling of pain [50]. The less educated person feels more pain than the educated person. But in our study, all of the participants were at least at their undergraduate level, educational qualification did not affect the pain or other physical discomforts, rather it varied according to vaccine types.”
Reviewer 3 Report
The authors made considerable improvements to the manuscript. My only doubt left is only about Figure 2, which in my opinion is now clearer than before but still a bit hard to appreciate - maybe the £D effect is not strictly necessary? I would like to ask what is the Editor's opinion on that
Author Response
We would like to convey our warm gratitude and thankfulness to the honourable reviewer for giving time and effort to review our manuscript.
In figure 2, we have removed the £D effect and made it easy to understand.